# Effects of *Coriandrum sativum* Seed Extract on Aging-Induced Memory Impairment in Samp8 Mice

**DOI:** 10.3390/nu12020455

**Published:** 2020-02-11

**Authors:** Yurina Mima, Nobuo Izumo, Jiun-Rong Chen, Suh-Ching Yang, Megumi Furukawa, Yasuo Watanabe

**Affiliations:** 1General Health Medical Center, Yokohama University of Pharmacy, Kanagawa 245-0066, Japan; 143067_br@yok.hamayaku.ac.jp (Y.M.); n.izumo@hamayaku.ac.jp (N.I.); megumi.furukawa@hamayaku.ac.jp (M.F.); 2School of Nutrition and Health Sciences, Taipei Medical University, Taipei 11031, Taiwan; jirchen@most.gov.tw; 3Research Center of Geriatric Nutrition, College of Nutrition, Taipei Medical University, Taipei 11031, Taiwan; 4Nutrition Research Center, Taipei Medical University Hospital, Taipei 11031, Taiwan

**Keywords:** *Coriandrum sativum* seed extract, memory impairment, senescence-accelerated mouse-prone 8 (SAMP8) mice

## Abstract

The purpose of this study was to investigate whether or not *Coriandrum sativum* seed extract (CSSE) can ameliorate memory impairment in senescence-accelerated mouse-prone 8 (SAMP8) mice. Sixteen 10-week-old male SAMP8 mice were divided into two groups, which were orally administrated water (SAMP8(−)) or CSSE (200 mg/kg/day; SAMP8(+)). Eight 10-week-old male Institute of Cancer Research (ICR) mice were used as a normal control group and were also orally administrated water. The mean escape time in the Barnes maze test of SAMP8(−) mice was significantly longer than that of ICR mice. However, SAMP8(+) mice showed a shorter mean escape time compared to that of SAMP8(−) mice. Neurofilament messenger (m)RNA levels significantly decreased in the frontal lobe of SAMP8(−) mice when compared with ICR mice, but significantly increased in SAMP8(+) mice relative to SAMP8(−) mice. In addition, mRNA levels of inducible nitric oxide synthase (iNOS) and neuronal (n)NOS significantly increased in the frontal lobe of SAMP8(−) mice, but only the mRNA level of nNOS significantly decreased in SAMP8(+) mice. These results indicated that continuous oral administration of CSSE for 12 weeks could ameliorate aging-induced memory declines in the senescence-accelerated SAMP8 mouse model.

## 1. Introduction

In 2017, the global population aged 60 years or over was 962 million and is expected to double by 2050 [1]. Aging societies exhibit changes in the leading causes of diseases and deaths. Many epidemiologic reports indicated that morbidity due to chronic and degenerative diseases increased in developing countries with high elderly populations [2]. Dementia is one of the age-induced degenerative diseases, and Alzheimer’s disease (AD) is the most common form of dementia [3]. The main symptoms of AD are loss of memory, reasoning, speech, and other cognitive functions [3]. The causes of AD are complicated, and the most credible theory is related to the accumulation of amyloid plaque and tau proteins in the brain [4]. Forster et al. also indicated that oxidative stress was a causal factor in brain senescence, involving oxidative molecular damage within different regions of the brain [5]. In addition, the brain is believed to be particularly vulnerable to oxidative stress due to a relatively high generation rate of oxygen-free radicals without commensurate levels of antioxidative defenses [6,7,8,9]. Based on these pathological mechanisms, new medical strategies are being developed for the therapy and prevention of AD, especially natural medicinal herbs, which have antioxidative properties, and some of them can be used as food.

*Coriandrum sativum* (CS) is utilized not only as a medicinal herb belonging to the Apiaceae (Umbellifera) family in Iran and Egypt, but also as an edible vegetable in Asian countries, such as Japan and Taiwan [10]. All parts of CS are edible, but the fresh leaves and dried seeds are the parts most often used in cooking [10]. Several animal studies have shown that CS has anti-diabetic [11], hypolipidemic [12], and anti-cancer effects [13] and sedative-hypnotic activity [14]. On the other hand, Sultana et al. summarized the literature on the antioxidative effects of spices and found that many spices, including CS, impart antioxidative effects in foods [15]. The report also states the in vivo antioxidant activities of coriander seed [16,17]. Moreover, our own study indicated that the leaf extract of CS has sedative effects due to hyperactivity of inhibitory neurons in the brain [18]. Thus, CS has not only peripheral actions, but also central action. These results indicate some ingredients of CS seed, and herbs can evaluate the cellular functions through the antioxidant activity and can pass through the blood-brain barrier. In interesting report of CS by Mani et al. showed that CS leaves can improve the memory deficits in mice [19].

Therefore, it was hypothesized that seeds of CS might also have a similar effect as leaves that could ameliorate degenerative brain diseases by decreasing oxidative stress caused by ageing. In this study, the senescence-accelerated mouse-prone 8 (SAMP8), which is a naturally occurring mouse line that displays a phenotype of accelerated aging, was used to explore this hypothesis.

## 2. Materials and Methods

### 2.1. Animals

A total of 8 10-week-old male Institute of Cancer Research (ICR) mice (Japan SLC, Shizuoka, Japan) and 16 10-week-old male SAMP8 mice (Japan SLC, Shizuoka, Japan) were used in this study. Mice were individually housed in temperature-controlled cages at 24 ± 1 °C and 55% humidity with a 12-h light-dark cycle. Mice were allowed free access to a standard rodent diet (CE2, Japan CLEA, Tokyo, Japan) and water during the entire experimental period. All procedures were approved by the Institutional Animal Care and Use Committee of Yokohama University of Pharmacy (2017-010).

### 2.2. Administration of the CSSE Extract

After acclimation for 1 week, SAMP8 mice were divided into two groups with (SAMP8(+) group) and without (SAMP8(−) group) oral administration of *Coriandrum sativum* seed extract (CSSE) (Sabinsa Japan, Tokyo, Japan) at random. The administration method for CSSE was according to Sukura et al. [18]. As in our previously reported paper and others, the dose-response (more than 100 mg/kg) of CS extract from seed and leaf showed the positive central and peripheral actions. Therefore, in this study, a 200 mg/kg CSSE was administered daily to mice in the SAMP8(+) group via oral gavage, whereas water was administered daily to mice in the SAMP8(−) group. Eight ICR mice were also administered water as the normal control group. The volume of oral administration was 10 mL/kg body weight (BW) of each mouse.

### 2.3. BW and Evaluation of Senescence

During the 12-week experimental period, the BW of each mouse was measured twice a week. Furthermore, the senescence of SAMP8 mice was evaluated according to a grading system comprising the following seven items in three categories: behaviors (reactivity and passivity), skin and hair (glossiness, coarseness, hair loss and ulcer), and spine (lordokyphosis) [20,21,22]. “Behaviors” and “skin and hair” scores were calculated by summing the scores of each item from 0 to 4. Moreover, the “spine” score was evaluated from 0 to 3 [23].

### 2.4. Barnes Maze Test

Two weeks before the end of the experiment, spatial learning and memory were evaluated using a Barnes maze test. The instrument (MB-10, Shinfactory Fukuoka, Japan) has a diameter of 455 mm, a height of 920 mm, and 20 holes with diameters of 25 mm, located 30 mm from the edge of the platform. The escape box has a width of 130 mm, a depth of 95 mm, and a height of 50 mm. The brightness of the maze was evenly set to 750 Lx. The maze was surrounded by partitions with three visible patterns on them, such as triangle, double-circle, and star patterns. After each mouse finished the test, the platform was cleaned with a 10% ethanol solution. To avoid visible and olfactory trails in the maze, the platform was turned once a day. The entire process was recorded by a digital hi-vision camera (HC-W580M, Panasonic, Tokyo, Japan).

#### 2.4.1. Spatial Memory Test

In the 10^th^ week, a repetitive learning test was performed. A mouse was placed in a cylinder (starter tube) with a diameter of 38 mm, a height of 73 mm, and a gray color. After 20 s, the starter tube was removed, and the mouse was given 2 min to search for the escape box. The escape time, i.e., the time to find the correct hole for the escape box, was measured. If the mouse did not find the escape box within 2 min, it was guided by a glass beaker and entered the escape box. After that, the mouse was kept in the escape box for 60 s before being returned to its cage. The test was executed four times a day for 4 days. If the mouse did not immediately enter the escape box after it had found the correct hole, 10 s was added to its time. The rate of change of the escape time was calculated on days 1, 2, and 4.

#### 2.4.2. Probe Trial

A probe trail was executed 2 weeks after the spatial memory test, which was also the day before sacrifice. The escape box was removed, and the test mouse was placed in the cylinder for 20 s. After 20 s had passed, the cylinder was removed, and the mouse was given 2 min to search for the target hole. The mouse was returned to its cage after the test. In the probe trial, the Barnes maze was divided into four zones. The mouse was placed in the maze within 60 s, and the time to discover each zone was recorded. The correct rate was calculated using the following formula:

Zone 4 (time in seconds or travel distance)/Zone (1+2+3+4) (time in seconds or travel distance).

### 2.5. Animal Sacrifice and Sample Collection

All mice were sacrificed at the end of week 12 of the experiment. Blood samples were collected in heparin-containing tubes and centrifuged (1200 × *g* for 15 min at 4 °C) to obtain plasma samples. Furthermore, brain tissues were rapidly excised and the frontal lobe and hippocampus were removed. All samples were stored at −80 °C for further measurements as described here.

#### 2.5.1. Plasma Superoxide Dismutase (SOD) Activity

A commercial kit (Dojindo Molecular Technologies, Rockville, MD, USA) was used to determine the activities of superoxide dismutase (SOD) in plasma as measured by the inhibition of formazan dye formation.

#### 2.5.2. Total Messenger (m)RNA Extraction and Real-Time Quantitative Polymerase Chain Reaction (QPCR)

Total RNA in the frontal lobe and hippocampus was extracted by the addition of POLYTRON PT 1300 D (Central Scientific Trade, Tokyo, Japan) with Isogen (Nippon Gene, Tokyo, Japan), according to the manufacturer’s instructions, with minor modifications as described in a previous study [11]. The RNA concentration was then standardized to 50 ng/μL, according to the Super Script VILO cDNA Synthesis kit protocol (Thermo Fisher Scientific, Waltham, MA, USA), and incubated at 25 °C for 10 min, 42 °C for 60 min, and at 85 °C for 5 min to obtain complementary (c)DNA. The resulting cDNA was amplified in a 96-well PCR plate with LightCycler 480II (Hoffmann-La Roche, Basel, Switzerland) on an Applied BioSystems 7300 Real-Time PCR System (Thermo Fisher Scientific, Tokyo, Japan). Gene levels were determined according to glyceraldehyde-3-phosphate dehydrogenase (GAPDH) as the internal standard. Information on primers is given in Table 1.

### 2.6. Statistical Analysis

All values are expressed as mean ± standard error of the mean (SEM). Statistical significance was assessed using a one-way analysis of variance (ANOVA), followed by Fisher’s protected least significant difference (PLSD) test, and was accepted at the *p* < 0.05 or *p* < 0.01 levels. Results were analyzed using Stat View 5.0 statistical software (SAS Institute, Cary, NC, USA).

## 3. Results

### 3.1. Final BW and Appearance Changes

Changes in the final BW and appearance in each group are shown in Table 2. The final BW had singnificantly increased in the SAMP8(+) group compared to the SAMP8(−) group, although these changes in SAMP8 groups were significantly lower than that of ICR mice. The appearances of senescence in the SAMP8(−) group significantly changed compared to those of the ICR group in behavoir, skin, and hair; however, changes seen in the SAMP8(+) group were almost the same as those of the SAMP8(−) group.

### 3.2. Performance on the Barnes Maze Test

#### 3.2.1. Spatial Memory Test

As shown in Figure 1, the escape time in the SAMP8(−) group was significantly longer than that of the ICR group on days 1, 2, and 4. However, the SAMP8(+) group showed a significantly shorter escape time on day 2 compared to the SAMP8(−) group.

#### 3.2.2. Probe Trial

The performance on the probe trial is shown in Figure 2. The escape time of SAMP8(−) mice was significantly longer than that of the ICR group. On the contrary, the escape time was significantly shortened in the SAMP8(+) group compared to the SAMP8(−) group.

### 3.3. Plasma SOD activity

SOD activity in each group is shown as Figure 3. Compared to the ICR group, the SAMP8(−) group had significantly higher plasma SOD activity. On the other hand, there was no difference in SOD activities between the SAMP8(−) and SAMP8(+) groups.

### 3.4. Gene mRNA Levels in the Frontal Lobe and Hippocampus

As shown in Figure 4a, in the frontal lobe, no difference was observed in messenger (m)RNA levels of apolipoprotein E (apoE) among all groups. However, neurofilament light (NF-L) mRNA was significantly lower in the SAMP8(−) group, whereas mRNA levels of neuronal nitric oxide synthase (nNOS) and inducible nitric oxide synthase (iNOS) were significantly higher compared to those of the ICR group. In addition, the NF-L mRNA level was siginificantly elevated in the SAMP8(+) group compared to the SAMP8(−) group (Figure 4a). Althought the mRNA levels of nNOS were significantly inhibited, the iNOS mRNA level only showed a reduced trend in the SAMP8(+) group (Figure 4a). In the hippocampus, only the NF-L mRNA level was significantly decreased in the SAMP8(−) group when compared to the ICR group; however, the SAMP8(+) group showed an increased trend relative to the SAMP8(−) group (Figure 4b).

## 4. Discussion

### 4.1. SAMP8 Mouse Model

SAMP8 mice have been widely used in aging research to study such phenotypes as immune dysfunction, osteoporosis, and brain atrophy. It has been verified that SAMP8 mouse exhibit moderate to severe degrees of activity loss, hair loss, and lordokyphosis, and also aging-associated behavioral impairments, including learning and memory difficulties [20,21,22,23]. The purpose of this experiment was to investigate the effects of CSSE on oxidative stress, memory impairment, and neuronal protection in SAMP8 mice. The SAMP8 mouse model, instead of the Alzheimer mouse, was used in this study to demonstrate early memory disorders caused by AD [24].

### 4.2. CSSE and Appearance Changes of Senescence

As shown in Table 2, BWs of the SAMP8 groups were significantly lower than that of the ICR group. Marie et al. reported that SAMP8 mice showed lower BWs, whereas CBA mice (normal mice from which the SAMP8 strain originated) retained stable BWs at the same age [25]. These BW changes can be explained by the decrease in muscle mass due to the advanced stage of the senescence process, as previously described by Guo et al. [26]. On the other hand, a different appearance was also observed in SAMP8(−) mice, including hair loss and lordokyphosis of the spine (Table 2). These aging-related appearance changes in this study were consistent with findings of Miyamoto et al. [27,28,29].

When SAMP8 mice were supplemented with CSSE (SAMP8(+) mice), BW reductions and appearance changes were also seen (Table 2). Treatment with CSSE could not protect the mice from physical decline.

### 4.3. CSSE and Brain Function

#### 4.3.1. Spatial Memory Test and Probe Trial

As shown as the result of the spatial memory test, mice in the SAMP8(−) group had longer escape times than ICR mice on days 1, 2, and 4, but the escape time was significantly shortened after CSSE supplementation (Figure 1). According to the daily change within each group, the escape time decreased gradually in ICR group (Figure 1). However, the escape time was decreased on day 3 in SAMP8(−) mice and on day 2 in SAMP8(+) mice, then maintained at the similar escape time until day 4. It seems that the SAMP8(−) was a slow learner for spatial training and might be improved by CSSE supplementation (Figure 1). Following sufficient acquisition training, the escape tunnel was removed and a probe trial was administered to assess the spatial reference memory. Results showed that mice in the SAMP8(−) group also spent more time before entering the target quadrant (Figure 2). In contrast, the escape time of the SAMP8(+) group significantly improved, not only in the spatial memory test, but also in the probe trial. These results indicated that SAMP8 mice with CSSE supplementation showed learning and memory impairments. The accumulation of cell damage due to oxidative stress and inflammation, which can cause aging, might have been reduced [30]. On the other hand, Deepa and Anuradha previously proved that CSSE has strong antioxidative properties [31]. Therefore, we speculated that CSSE might have the potential for delaying senescence, since it reduces oxidative stress.

#### 4.3.2. mRNA Levels of Related Protein Biomarkers in the Brain

Memory retention is affected by synaptic plasticity, which is related to a variety of proteins, such as apoE, and NF-L [32,33,34,35], and by a variety of factors, such as amyloid-β (Aβ) accumulation and oxidative stress, including iNOS and nNOS [36]. In this study, differences in apoE mRNA levels were not observed among the three groups in the frontal lobe or in the hippocampus (Figure 4a,b). Cholesterol released from apoE-containing lipoprotein particles is used to support synaptogenesis and maintain synaptic connections [37]. However, the effects of apoE-containing lipoproteins in supporting synaptogenesis and maintenance of synaptic connections in vivo are still inconsistent [30]. Evans suggested that apoE isoforms play key roles in the development of AD via their effects on Aβ aggregation and clearance [38]. Therefore, it is necessary to examine the various apoE isoforms in future studies.

The damage of hippocampus and frontal lobes is the risk factor of AD, since these two parts of the brain contribute to episodic memory performance. In this study, the NF-L mRNA levels in the frontal lobe and hippocampus were significantly lower in SAMP8 mice without treatment (Figure 4a). Together with the medium and heavy subunits, NF-L is one of the cytoskeletal proteins in neurons and is released into the extracellular space after neuronal damage occurs [39,40]. Disanto et al. confirmed that plasma NF-L concentrations in patients with AD were significantly higher compared to those of healthy controls, which might be considered a plasma biomarker for the screening of neurodegeneration in AD [40]. On the other hand, a significant decrease in NF-L in the occipital cortex of AD patients was also found, which may reflect neuronal loss [41]. These previous findings support our notion that SAMP8 mice administrated only water may have neuronal damage, according the lower NF-L mRNA level in the frontal lobe and hippocampus. However, the protein levels of NF-L in plasma and brains definitely have to be measured in future studies. When SAMP8 mice were administrated CSSE for 12 weeks, the NF-L mRNA level in the frontal lobe was significantly elevated, which might indicate neuronal impairment.

#### 4.3.3. Oxidative Stress

In this study, mice in the SAMP8(−) group showed significantly higher mRNA levels of nNOS in the frontal lobe compared to ICR mice (Figure 4a). Overproduction of nitric oxide (NO) by nNOS or iNOS is one of the fundamental causes underlying neurodegenerative disorders and neuropathic pain [42]. Excess NO itself can impair cellular energy production via interacting with iron-sulfur centers in the mitochondrial electron transport chain [42]. NO can also produce reactive nitrogen species (RNS) and reactive oxygen species (ROS) [43]. Therefore, it is suggested that SAMP8 mice without supplementation showing learning and memory impairment may have been due to the excess NO level produced by higher mRNA levels of nNOS or iNOS. After supplementation with CSS, SAMP8 mice showed significantly lower mRNA levels of nNOS in the frontal lobe; this means that CSSE might reduce the production of RNS and ROS and thus ameliorate oxidative stress induced by natural aging (Figure 4a). In this study, the antioxidative stress potential of CSSE was also supported by another biomarker, since the plasma SOD activity in the SAMP8(+) group was almost the same as that seen in the ICR group (Figure 3). In contrast, the plasma SOD activity in SAMP8(−) mice was much higher than that of both the SAMP8(+) and ICR groups. These results showed that the production of several kinds of superoxide are increased by aging, and repeated treatment with CSSE reduced such production.

On the other hand, in this study, changes in related biomarkers were found in the frontal lobe, but not in the hippocampus. The brain is divided into areas that are each responsible for different areas of functioning. The frontal lobes are involved in executive function, memory, and social behavior, while the hippocampus is mainly associated with memory, particularly long-term memory, which plays an important role in spatial navigation. Miyamoto et al. indicated that significant hippocampus-related memory impairment was observed in SAMP8 mice after 12 months (48 weeks old) of age [12]. In this study, we observed SAMP mice from 12 to 24 weeks of age, which is the early stage of brain damage. That is the reason explaining why there were no changes in related biomarkers in the hippocampus of SAMP8 mice.

### 4.4. Dosage and Effective Components of CSSE

Patel et al. indicated that CSSE is nontoxic at up to 3000 mg/kg BW and can be considered safe for consumption [44]. Previous studies confirmed that CSSE significantly increased pentobarbital sodium-induced sleeping times at a dose of 200 mg/kg BW [45,46]. The main ingredient of coriander seed, linalool, was verified to exert a sedative effect in mice [47]. Latha et al. also described how the anxiolytic activity of CSSE occurred at doses of 50, 100, and 200 mg/kg BW in mice, and the above effects may have been due to the presence of sterols, tannins, and flavonoids in the extract [48]. Those components have the potential to activate inhibitory neurons and reduce oxidative stress [10,45,46,47,48]. Based on previous research, SAMP8 mice were orally administrated 200 mg/kg BW of CSSE in this study, which improved the spatial memory via increasing the mRNA level of NF-L and decreasing the mRNA level of nNOS in the frontal lobe of the brain.

### 4.5. Limitations and Future Research

There are some limitations in this study. First, only using Barnes maze test is not sufficient to validate the behavioral outcome in this study. More animal behavior trials are necessary. However, although many behavioral tests for evaluating the anti-amnesia have been reported, most of these tests must be considered other factors, such as stress, smell, anxiety, and fear. In most case of early senile dementia, the loss of spatial memory can appear. Therefore, the Barnes maze test is suitable for seeking the early senile dementia, at present. Second, more biomarkers should be measured (e.g., apoE isoforms), as well as antioxidative-related biomarkers in the brain and plasma (e.g., the protein level of antioxidative enzymes). Third, SAMP8 mice were still in the early stage at the end of the experiment, which may have resulted in inconspicuous improvements in hippocampal damage by CSSE. Extension of the experimental period is necessary in future studies. Additionally, further studies also need to separate and identify the active components in *Coriandrum sativum* seed extract and their effects on memory impairment in SAMP8 mice.

## 5. Conclusions

This study demonstrated that administration of *Coriandrum sativum* seed extract (200 mg/kg BW) exhibited ameliorative effects on memory impairment in SAMP8 mice. We speculated that the mechanisms involved in repairing memory deficits by *Coriandrum sativum* seed extract were associated with increased NF-L mRNA and decreased nNOS mRNA in the frontal lobe of the brain.

## Figures and Tables

**Figure 1 nutrients-12-00455-f001:**
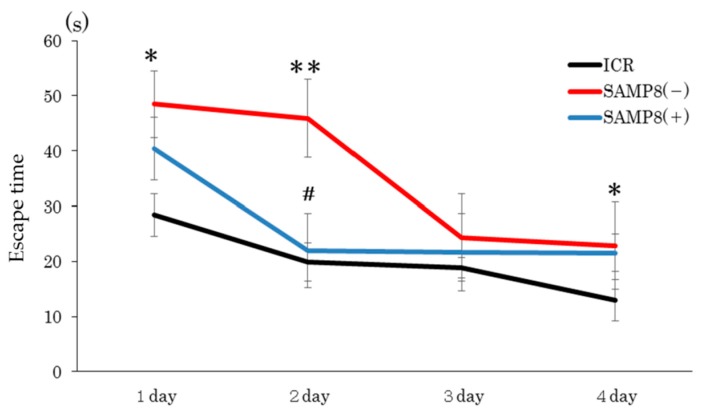
Effects of *Coriandrum sativum* seed extract on the performance of a spatial memory test in ICR and SAMP8 mice. Values are expressed as the mean ± SEM. An asterisk indicates a significant difference compared to the ICR group (* *p* < 0.05; *** p* < 0.01). A hash mark (#) indicates a significant difference compared to the SAMP8(−) group. ICR, Institute of Cancer Research; SAMP8(−), SAMP8 mice without *C. sativum* supplementation; SAMP8(+), SAMP8 mice with *C. sativum* supplementation.

**Figure 2 nutrients-12-00455-f002:**
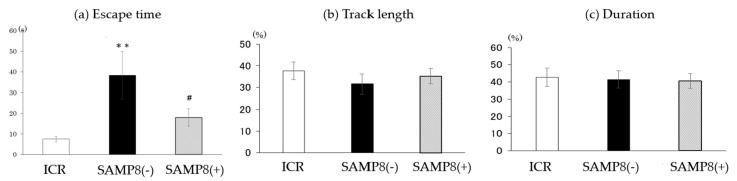
Effects of *Coriandrum sativum* seed extract on the performance on the probe trial by ICR and SAMP8 mice. Values are expressed as the mean ± SEM. (**a**) Time spent in the target quadrant; (**b**) percent of the exploring distance in the target quadrant; (**c**) percent of the exploring time spent in the target quadrant. An asterisk indicates a significant difference compared to the ICR group (*** p* < 0.01). A hash mark (#) indicates a significant difference compared to SAMP8(−) mice. ICR, Institute of Cancer Research; SAMP8(−), SAMP8 mice without *C. sativum* supplementation; SAMP8(+), SAMP8 mice with *C. sativum* supplementation.

**Figure 3 nutrients-12-00455-f003:**
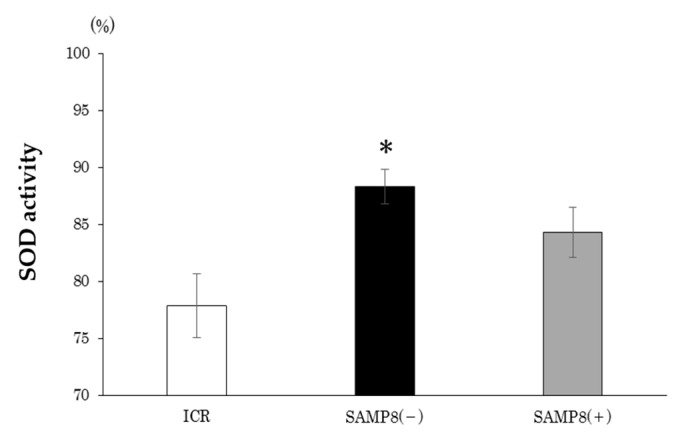
Effects of *Coriandrum sativum* seed extract on plasma superoxide dismutase (SOD) activities of ICR and SAMP8 mice. Values are expressed as the mean ± SEM. An asterisk indicates a significant difference compared to the ICR group (** p* < 0.05). ICR, Institute of Cancer Research; SAMP8(−), SAMP8 mice without *C. sativum* supplementation; SAMP8(+), SAMP8 mice with *C. sativum* supplementation.

**Figure 4 nutrients-12-00455-f004:**
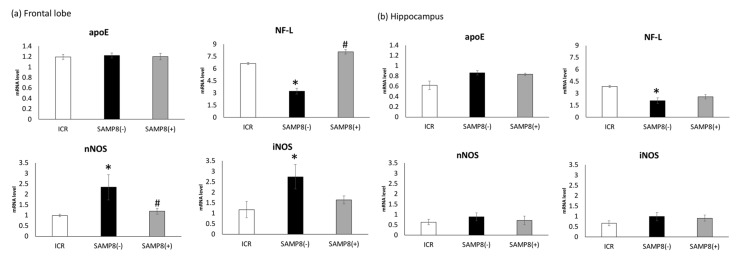
Effects of *Coriandrum sativum* seed extract on messenger (m)RNA expression levels of apolipoprotein E (apoE), neurofilament light (NF-L), inducible nitric oxide synthase (iNOS), and neuronal (n)NOS in the frontal lobe and hippocampus of ICR and SAMP8 mice. Values are expressed as the mean ± SEM. An asterisk indicates a significant difference compared to the ICR group (** p* < 0.05). A hash mark (#) indicates a significant difference compared to SAMP8(−) mice. ICR, Institute of Cancer Research; SAMP8(−), SAMP8 mice without *C. sativum* supplementation; SAMP8(+), SAMP8 mice with *C. sativum* supplementation.

**Table 1 nutrients-12-00455-t001:** Primers used for the quantitative polymerase chain reaction.

Gene	Forward 5’→3’	Reverse 5’→3’	Universal Probe Library
GAPDH	AGCTTGTCATCAACGGGAAG	TTTGATGTTAGTGGGGTCTCG	#9
apoE	TAGAGGAGGTGCGTGAGCA	ATTTGCTGGGTCTGTTCCTC	#25
NF-L	GGAAAGAGCCGAGCAGACATC	GCCGTTCTGCTTACAGTGG	#17
nNOS	GGCGTTCGTGATTACTGTGA	TCTTCCTCATGTCCAAATCCA	#69
iNOS	CTTTGCCACGGACGAGAC	TCATTGTACTCTGAGGGCTGAC	#13

GAPDH, glyceraldehyde-3-phosphate dehydrogenase; apoE, apolipoprotein E; NF-L, neurofilament light; nNOS, neuronal nitric oxide synthase; iNOS, inducible nitric oxide synthase; #, probe number of Universal Probe Library.

**Table 2 nutrients-12-00455-t002:** Effects of *Coriandrum sativum* seed extract on the final body weight and appearance change in ICR and senescence-accelerated mouse-prone 8 (SAMP8) mice.

	ICR	SAMP8(−)	SAMP8(+)
Final Body Weight (g)	45.2 ± 0.9	26.0 ± 0.6#	27.5 ± 0.5 *
Behavior	1.1 ± 0.1	3.5 ± 0.5#	3.4 ± 0.4
Skin and Hair	3.9 ± 0.9	6.8 ± 0.9#	7.3 ± 0.8 #
Spine (Lordokyphosis)	0.0 ± 0.0	0.0 ± 0.0	0.0 ± 0.0

Values are expressed as the mean ± standard error of the mean (SEM). An asterisk (*) indicates a significant difference compared to the SAMP8(−) group (*p* < 0.05). A hash mark (#) indicates a significant difference compared to ICR mice (*p* < 0.05). ICR, Institute of Cancer Research; SAMP8(−), SAMP8 mice without *C. sativum* supplementation; SAMP8(+), SAMP8 mice with *C. sativum* supplementation.

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
