# Peer review of "Effects of Coriandrum sativum Seed Extract on Aging-Induced Memory Impairment in Samp8 Mice"

_nutrients, 2020, doi:10.3390/nu12020455_

Round 1

Reviewer 1 Report

Overall, I think that this was an interesting paper suggesting that CSSE may have an impact on reducing the effects of senescence in mice. The paper was short and relatively well-written, although I recommend some additional explanation and information in some sections. In particular, I think that the basis for the use of CSSE needs further background and some of the tests will require additional description. 

Lines 19-20. These effects were significant, but were they significantly different from each other or from the control, or both? Explain.

End of introduction. The logical transition between the ultimate and penultimate paragraphs of the introduction requires more explanation. The reader needs more background as to why CSSE would be useful, including some discussion of previous work on the subject with that plant. I think that this sort of information would put the authors’ work in a better context and make their findings more appreciated.

Line 69-71. Please describe the grading system here. The authors reference previous papers, but spend a great deal of time in the results and discussion emphasizing these data. I think that the reader should know more about the tests and potential variation.

Line 95. Should be “trial” not “trail”.

Table 2. Please explain more about the measures.

Lines 174-175. Was there a significant difference between the SAMP8(-) and SAMP8(+)?

Figure 4b. The hippocampal NF-L scores seem significantly different. While I cannot see the numbers, with the same sample sizes and mean/SEM, smaller differences were considered significant. So, I am confused. I would recommend that the authors include actual p-values and stat scores. If the journal will not allow that, then I suggest at the least discussing the effect in the results and discussion, especially in contrast to the NF-L measures in the frontal lobe.

Line 203. Please provide references for this statement.

Line 233. I think that the conclusion here that the CSSE delays senescence is written too strongly. I suggest tempering that conclusion.

Lines 247-258. I suggest checking the statistics on the hippocampal NF-L and/or discussing the effects here.

Nice study.

Author Response

Dear Professor:

Thank you for your kindness and patience. We added some additional explanation and revised the manuscript in accordance with your advice as the attached file. We hope you find our responses satisfactory and that the manuscript is now acceptable for publication.

Reviewer 2 Report

The manuscript by Mima et al., describes the impact of coriandrum sativum (CS) on the improvement of memory impairment in aged SAMP8 mice.  Extracts from the seed of CS, showed an attenuation of the memory impairment seen in the SAMP8 mice with age.  Changes were seen in other markers in the brain and blood of the mice, suggesting an alteration in the aging process.  The study is reasonably well done, but there are some issues that need to be addressed.  

1) In Figure 1, the SAMP8 mice without the CS, seen to show a similar profile of performance in the spatial memory task at 3-4 days of training.  Why?

2) In Figure 2, why is the % duration time of the time in the target quadrant the same for all groups?

3) Figure 3 shows the SOD activity in the blood.  Can you see an increase in the protein levels in the blood?

4) In Figure 4, do the protein levels correlate with the mRNA levels for the various genes?  This is important to show a relationship to functional changes. 

5) The Barnes Maze test is good, but one single test is really not sufficient to validate the behavioral outcomes.

6) Eight mice per group is not sufficient to validate the results.  10-12 are really needed.  

7) Were the mice randomized to the groups, and were the experimenters blinded to the test groups?

8) Why was the 200 mg/kg dose chosen?

Author Response

(The authors gave the same response as above.)
